# Nano-Scale Engineering of Heterojunction for Alkaline Water Electrolysis

**DOI:** 10.3390/ma17010199

**Published:** 2023-12-29

**Authors:** Yao Chen, Zhenbo Xu, George Zheng Chen

**Affiliations:** 1The State Key Laboratory of Refractories and Metallurgy, Faculty of Materials, Wuhan University of Science and Technology, Wuhan 430081, China; 2Department of Chemical and Environmental Engineering, Faculty of Engineering, University of Nottingham, Nottingham NG2 7RD, UK

**Keywords:** heterojunction, alkaline water electrolysis, HER, OER, d-band center, scaling relation

## Abstract

Alkaline water electrolysis is promising for low-cost and scalable hydrogen production. Renewable energy-driven alkaline water electrolysis requires highly effective electrocatalysts for the hydrogen evolution reaction (HER) and the oxygen evolution reaction (OER). However, the most active electrocatalysts show orders of magnitude lower performance in alkaline electrolytes than that in acidic ones. To improve such catalysts, heterojunction engineering has been exploited as the most efficient strategy to overcome the activity limitations of the single component in the catalyst. In this review, the basic knowledge of alkaline water electrolysis and the catalytic mechanisms of heterojunctions are introduced. In the HER mechanisms, the ensemble effect emphasizes the multi-sites of different components to accelerate the various intermedium reactions, while the electronic effect refers to the d-band center theory associated with the adsorption and desorption energies of the intermediate products and catalyst. For the OER with multi-electron transfer, a scaling relation was established: the free energy difference between HOO* and HO* is 3.2 eV, which can be overcome by electrocatalysts with heterojunctions. The development of electrocatalysts with heterojunctions are summarized. Typically, Ni(OH)_2_/Pt, Ni/NiN_3_ and MoP/MoS_2_ are HER electrocatalysts, while Ir/Co(OH)_2_, NiFe(OH)_x_/FeS and Co_9_S_8_/Ni_3_S_2_ are OER ones. Last but not the least, the trend of future research is discussed, from an industry perspective, in terms of decreasing the number of noble metals, achieving more stable heterojunctions for longer service, adopting new craft technologies such as 3D printing and exploring revolutionary alternate alkaline water electrolysis.

## 1. Introduction

With the emission of greenhouse gases and pollutants from globally consumed fossil energy, it is urgent to develop an alternative sustainable clean energy source. Hydrogen is a clean energy carrier with zero greenhouse gas emissions and with a high specific energy of 120 MJ kg^−1^, hence the hydrogen economy was put forward in the 1970s [1]. The industrialized production of hydrogen mainly includes steam methane reforming, coal gasification and water electrolysis [2,3]. The first two account for more than 95% of the global hydrogen fuel production [4], but they generate CO_2_ by high-temperature reactions [5]. Nevertheless, hydrogen production via the electrolysis of water has the advantages of the high purity of produced hydrogen, no greenhouse gas generation and a fast production rate [6]. Renewable energy sources, such as solar energy and wind energy, exhibit intermittent power generation, making them challenging to connect to the grid. The electrolysis of water driven by electric energy generated from these renewable but intermittent energy sources can improve the utilization rate of renewable energy and produce high-pressure hydrogen efficiently and cleanly [7].

In the standard state at 298 K, the Gibbs energy change is 237.1 kJ for splitting 1 mol of water into hydrogen and oxygen gases, as shown in Equation (1). According to Faraday’s law, it takes a theoretical voltage of 1.23 V [8].
(1)$$H_{2}O=H_{2}↑+\;\frac{1}{2}O_{2}↑\left({∆G}^{o}=+237.1\;kJ\;{mol}^{-1}\right)$$

However, practical application requires additional potential to overcome the reaction resistance, external circuit resistance, electrolyte resistance and membrane resistance.

Reaction resistance is the inherent resistance of electrochemical reactions involved in the HER and OER, that is, the reaction overpotential depends on the surface activity of electrode materials. Low overpotential can reduce energy consumption and improve energy efficiency, so it is imperative to find high-activity, low-cost and stable catalysts for water electrolysis.

The electrolysis of water consists of two half reactions, namely a cathodic HER and an anodic OER [9]. Noble metal nanoparticles show the outstanding catalytic activities in H_2_ generation [10,11,12]. Similarly, noble metals also demonstrate excellent electrocatalytic performances in the HER. Furthermore, noble metal catalysts can resist the strong corrosiveness of acidic electrolytes. To date, Pt is the best catalyst for the HER, while RuO_2_ and IrO_2_ are often used for the OER [13]. Notably, S-doped SrIrO_3_ and IrO_x_/SrIrO_3_ still need overpotentials of 0.23–0.27 V to release hydrogen in an acidic electrolyte [14,15]. They all have low overpotentials, excellent catalytic performance and durability. However, the low reserves and high cost of noble metal materials limit the large-scale application of commercial electrolyzers [16]. To avoid these drawbacks of noble metals in the electrolysis of acidic water, current research primarily focuses on the electrolysis of alkaline water [17]. In an alkaline medium, the two-electron transferring HER is shown in Equation (2), divided into Volmer-Tafel, corresponding to Equations (3) and (4), and Volmer-Heyrovsky, corresponding to Equations (3) and (5), mechanisms [18,19]. Specifically, the first step of the HER is the Volmer reaction, in which water molecules absorbed on the active sites get electrons to be reduced to adsorbed hydrogen (H*) and release OH^−^. When the coverage of H* on active sites is high enough, the Tafel reaction occurs, in which two H* forms H_2_. Otherwise, the Heyrovsky reaction occurs, in which an H* combines water and electrons to form H_2_.
(2)$$2H_{2}O+2e=H_{2}↑+\;{2OH}^{-}$$
(3)$$\mathrm{Volmer}:H_{2}O+e+*=H^{*}+{OH}^{-}$$
(4)$$Tafel:{2H}^{*}=H_{2}↑+\;2*$$
(5)$$Heyrovsky:H^{*}+H_{2}O+e=H_{2}↑+\;{OH}^{-}+*$$
where * refers to an empty active site and H* denotes a hydrogen atom bonded to an active site. In contrast, the OER is a complex four-electron reaction as shown in Equation (6), during which three oxygen intermediates (HO*, O* and HOO*) form before producing O_2_. The widely accepted mechanism of OER in alkaline electrolytes is as shown in Equations (7)–(10) [20].
(6)$$4{OH}^{-}-4e=O_{2}+{2H}_{2}O$$
(7)$${OH}^{-}+*-e={HO}^{*}$$
(8)$${HO}^{*}+{OH}^{-}-e=H_{2}O+O^{*}$$
(9)$${OH}^{-}+O^{*}-e={HOO}^{*}$$
(10)$${OH}^{-}+{HOO}^{*}-e=O_{2}+H_{2}O+*$$

Cheap transition metal compounds, such as hydroxides, double hydroxides, oxides, phosphides, sulfides, nitrides, carbides, selenides and tellurides, have been employed as catalysts in the electrolysis of alkaline water [21]. However, single-component materials also face some problems, such as high overpotential, insufficient activity and poor durability, so defect engineering, heteroatom doping, metal alloying and heterojunction engineering are often employed to further increase active sites and reduce overpotential [22].

As we know, it is proposed by Sabatier that an ideal catalyst must bind to the reactant at an intermediate strength which is neither too weak nor too strong. If the bond is too weak, the catalyst and the reactant will hardly interact with each other; whereas if the bond is too strong, the reactant will not desorb from the catalyst surface, effectively inhibiting further reactions [23,24]. A heterostructure has a multi-component synergistic effect, typically an electronic effect and an ensemble effect, which can significantly improve the adsorption free energy of catalysts with intermediates. Furthermore, the favorable heterogeneous junctions can optimize the reaction path and enhance the reaction kinetics [25]. This provides a rational method for designing electrocatalysts based on the composition and characteristics of heterogeneous junctions. For example, the Pt-Ni(OH)_2_ heterostructure, composed of Ni(OH)_2_ nanoclusters on the surface of a platinum electrode, showed increased catalytic activity for the HER by eight times. The edges of Ni(OH)_2_ clusters promoted the dissociation of water and the formation of hydrogen intermediates [26].

In this review, the mechanism of the heterogeneous junctions for the HER and OER in the electrolysis of alkaline water was firstly illuminated. Then the preparation and catalytic performance of representative heterogeneous catalysts were introduced. Finally, the trend of heterogeneous junction engineering development was discussed.

## 2. Mechanisms

### 2.1. Heterojunction Characteristics

A heterojunction is defined as the interface field formed by the contact of two different semiconductors. Nowadays, heterojunctions have extended from semiconductor-semiconductor (p-n) heterojunctions to metal-semiconductor (Schottky) heterojunctions and atypical semiconductor and ionic conductor heterostructures [27]. In the field of electrocatalysis, heterogeneous junction catalysts are defined as catalysts containing heterogeneous or multicomponent interfaces. Figure 1 shows that heterojunctions are divided into planar heterojunctions (Type I), lateral heterojunctions (Type II) and hierarchical (vertical) heterojunctions (Type III) [25]. Planar and hierarchical interfaces are popularly designed in the electrocatalytic field due to the small contact area between different components, facilitating the maximization of adsorption sites. The lateral interface, with a larger contact area, shows much stronger electron and energy band structure interaction, which is widely applied in the photocatalysis reactions.

When two (or more) phases come into contact to form an interface, lattice mismatch or lattice distortion typically occur due to the diversity of lattice parameters. The mismatched interface contains many dangling bonds, exhibiting more defects and yielding lattice strain. An interdiffusion region, akin to the depletion layer in semiconductors, would realize the combination and stabilization of multiple components. Highly lattice-mismatched interfaces with abundant bond distortions are usually highly active for catalysis. The lattice mismatch results in distinctive atomic coordination, exposing more active sites and thereby improving catalytic performance. Peculiarly, when the lattice is matched, the interface will show a coherently crystalline matrix without dangling bonds, which can be considered as a boundary to separate different components with a continuous bandgap grading. The chemical bonds in the lattice-matched interface are strong and stable. A notable example is the lattice-matched Mo_2_C-Mo_2_N heterostructure prepared by the gradient heating epitaxial growth method, which exhibits near-zero hydrogen adsorption free energy and facilitates water dissociation [28].

Due to the difference in electronic structure or bond energy, the heterogeneous junction exhibits a change in the electronic density at the interface of the two phases, thereby altering the adsorption and desorption energies of intermediates and catalysts [29]. As two phases exist at the heterogeneous interface at the same time, the adsorption of various intermediates on the heterogeneous junction is synergistically optimized to greatly accelerate reaction kinetics. For example, O* was adsorbed on IrO_x_ and OH* on Ni(OH)_2_ simultaneously at the IrO_x_/Ni(OH)_2_ interface, the combination of these two intermediates at the interface accelerates the formation of OOH* [30]. These characteristics of heterogeneous junctions are attributed to two classical effects: the electronic effect and ensemble effect [31].

### 2.2. Two Classic Effects of Heterojunction

#### 2.2.1. Ensemble Effect

The ensemble effect involves determining the number of surface atoms necessary to form the chemisorption bonds to the surface of reactants, intermediates and transition states. The concept referring to the randomly alloyed geometries about the ensemble of a certain number of atoms was introduced in heterogeneous catalysis to explain the improved activity and/or selectivity of bimetallic catalysts compared with pure metallic catalysts [32]. Subsequently, the ensemble requirement is defined as the minimum number of metal atoms and the corresponding spatial arrangement needed to catalyze a specific reaction most efficiently. It can also evaluate the pure metal single-crystal and even single-atom catalysts [33]. For heterojunction catalysts in electrocatalysis, the ensemble effect is defined as a synergistic effect on the catalytic process by the modification of the surface in each side of the heterointerface [31]. The two phases in heterojunctions play important roles simultaneously in the catalytic reaction. The ensemble effect depends on bridge- or interconnected adsorption sites with different coordination environments in the multi-component heterostructures to optimize and accelerate the adsorption process of different reaction intermediates. It can potentially integrate the active sites of different components to form the multi-site reaction pathway.

For instance, the significantly increased electrocatalytic activity of the Ni(OH)_2_ on the Pt(111) surface for the HER, compared with that of Pt(111), is attributed to the ensemble effect. Ni(OH)_2,_ with a strong electrostatic affinity, can preferentially promote the attraction and dissociation of water, while the adjacent Pt sites with an appropriate adsorption energy can accelerate the generation of H* [26]. Another example is Ir nanoparticles on Ni(OH)_2_ for the OER [30]. An in situ Ir/Ni(OH)_2_ heterojunction formed the heterointerface of IrO_x_ and NiOOH during the OER, which were the actual active sites. The adsorption of OER intermediates at the interface was no longer limited to single-component catalysts, so OH* and O* substances were adsorbed on NiOOH and IrO_x_, respectively. The two intermediates accelerated the formation of OOH* at the interface of the heterostructure, ultimately breaking the scaling relation, which will be discussed in the next section, and greatly improving the kinetics of the OER. Therefore, a rational design of heterostructures based on the ensemble effect can significantly enhance the catalytic performance and even surpass the activity limits of catalysts for the HER and OER.

#### 2.2.2. Electronic Effect

The electron effect arising from the energy band difference reflects the electronic interaction between the multiple components which constitute the heterojunction. Within a single catalyst, both electron-rich and electron-deficient states coexist, leading to oriented electron transportation and creating a charge distribution gradient at the interface. The electron effect can regulate the adsorption energies of reaction intermediates by coupling the electronic configurations of the catalyst. In our opinion, the electron effect lies in the enhanced kinetics in terms of microscopic changes in electronic configurations due to heterojunctions, while the ensemble effect lies in terms of the intuitive synergistic effect where multiple components increase the kinetics of different elemental steps.

The electrocatalytic activity of the catalyst largely depends on the adsorption energy and desorption energy of the reactive species. In the processes of both the HER and OER, the catalytic reaction involves the adsorption of reactants, the formation and transformation of intermediates and the desorption of products on the catalyst surface. The adsorption of intermediates is a crucial step in the electrocatalytic reaction and can be categorized into physical adsorption and chemical adsorption. Physical adsorption relies on physical forces, such as van der Waals forces or electrostatic attraction, forming attractive forces without the formation of chemical bonds. Physical adsorption is often associated with material properties such as specific surface area and pore size. On the other hand, chemical adsorption results from the force of chemical bonds between the adsorbent and adsorbate. For electrocatalytic reactions, chemical adsorption is usually considered, because the intermediate forms a chemical bond with the catalyst.

Adsorption and desorption energies of the intermediate and transition metal-based catalyst are associated with the d-band center theory. The valence states of the transition metal surface atoms can be divided into the free-electron-like 4 s electron states and the more localized 3d electron states. The energy range of metal s states is very wide, while the range of d states is generally near the Fermi level, corresponding to the red part in Figure 2a [34]. In the process of forming chemical bonds between the adsorbate and transition metal, the transition and broadening of the s and d states of the transition metal and the valence states of the adsorbate occur on the surface of the transition metal. Firstly, the coupling between the adsorbate and the s band of the metal occurs to form the renormalized s band. Then the reformed s band couples to the d band, resulting in bonding states and anti-bonding states. The bond strength is inversely proportional to the filling of the anti-bonding states. Because the anti-bonding states are consistently higher than the d states, the center of the d band relative to the Fermi level is a crucial descriptor of the bonding strength between the adsorbate and the catalyst. In other words, relative to the Fermi level, the higher the energy of d states, the less the anti-bonding states are filled with electrons, leading to the stronger adsorption of the transition metal and adsorbate [35]. However, with the development of d-band center theory, in addition to the d state center relative to Fermi energy level, the shape of bonding and antibonding orbitals should also be considered. Figure 2b illustrates that the binding energies of C, N and O on Pd and Pd/X (X refers to late transition metals) alloys exhibit a V-shaped relationship with *ε*_d_ (*ε*_d_ is the d-band center, i.e., the average energy of electronic d states projected onto a surface metal atom), which is a descriptor within the theoretical framework of the d-band model of surface chemisorption. Alloy systems located in the left region align with the general prediction of the d-band model. Outliers emerge in the right region when Pd is alloyed with Ag or Au. Although the *ε*_d_ of the surface Pd, marked as vertical lines in Figure 2c, shifts up slightly to Fermi level when alloying with Ag, the coupled antibonding states move down toward the Fermi level and become more occupied, weakening the C–Pd bond. The formation of electronic states away from *ε*_d_ is mainly due to the energy misalignment of interacting d orbitals, as shown in Figure 2d [36]. For example, when compared with pure NiSe_2_ or NiMoN, the NiSe_2_/NiMoN heterostructure exhibited a higher density of states (DOS) beyond Fermi level. The adsorption energy ΔG_H*_ of NiSe_2_/NiMoN was expected to be the lowest. However, in reality, it turned out to be the highest and was close to 0 [37].

The heterogeneous junction not only regulates the adsorption energy of adsorbates, but also the electron density at the interface of catalysts. Different work functions from two or more phases can alter the electron density at the interface, exerting a significant influence on the catalytic performance [29]. Figure 3a,b illustrates the change in electron density before and after the construction of the Mott-Schottky heterojunction between metallic NiS and n-type semiconductor MoS_2_ [38]. The NiS/MoS_2_ on carbon clothes delivered an overpotential of 87 mV at −10 mA cm^−2^, which was superior to 148 mV for MoS_2_ and 294 mV for NiS/MoS_2_ on carbon clothes. The work function represents the energy required for electrons transferring from the Fermi level to vacuum. Generally, the work function of metals is greater than that of N-type semiconductors. Before two-phase contact, the Fermi level of the metal is lower than that of the semiconductor. After contact, electrons spontaneously diffuse and transfer from the semiconductor to the metal, causing the energy band at the heterojunction surface to bend until the heterojunction reaches a state of thermal equilibrium, where the Fermi levels of the two phases are at the same level [39]. Once equilibrium is reached, a built-in electric field is generated, leading to redistributed electron cloud density and the formation of stable local nucleophilic/electrophilic regions [40]. The surface of an electron-rich catalyst with good electron-donating ability is suitable for reduction reactions such as the HER, while the surface of electron-deficient catalyst with good electron-accepting ability is suitable for oxidation reactions such as the OER. For example, in Figure 3c, the charge accumulation (magenta) and depletion (green) regions in the Ru/Ru_2_P heterojunctions are illustrated. Therefore, the modulation of electron density in heterogeneous junctions will also affect the electrocatalytic activity.

### 2.3. Scaling Relation

The OER overpotential is typically much higher than the HER overpotential. It is of great significance to reduce the OER overpotential by reasonably designing the catalyst. The catalytic performance is estimated by the magnitude of the potential-determining step for the OER. The free energy of the OER (Δ*G**_OER_*) is the maximum of those of the four steps from Equations (7)–(10), determined by the O* adsorption energy. This implies that either Equation (8) or (9) serves as the potential-determining step. Hence Δ*G**_OER_* can be expressed by Equation (11). The theoretical OER overpotential (*η**_OER_*), independent of pH, at standard conditions is given by Equation (12).
(11)$${\Delta G}_{OER}=\max\left[\left({\Delta G}_{O^{*}}^{o}-{\Delta G}_{{HO}^{*}}^{o}\right),\left({\Delta G}_{{HOO}^{*}}^{o}-{\Delta G}_{O^{*}}^{o}\right)\right]$$
(12)$$\eta_{OER}=\frac{{\Delta G}_{OER}}{e}-1.23$$

The energy diagram for the ideal OER catalyst is shown in Figure 4a. This ideal catalyst should be able to facilitate water oxidation just above the equilibrium potential, requiring all four immediate steps to have reaction free energies of the same 1.23 eV, numerically equivalent to the equilibrium potential 1.23 V. Therefore, the free energy difference between HOO* and HO* is 2.46 eV. Nevertheless, real catalysts do not exhibit this behavior. It was found that the energy levels of O*, HO* and HOO* intermediates in three perovskite materials with different bonding strengths moved together. A universal scaling relationship between the binding energy of HOO* and HO* was identified for various oxide surfaces, including perovskites, spinels, rutiles and rock salts, as shown in Equation (13). The free energy difference between HOO* and HO* is 3.2 ± 0.2 eV (Figure 4b). Combining Equations (11)–(13), *η**_OER_* can be expressed by Equation (14) [42].
(13)$${∆G}_{{HOO}^{*}}-{∆G}_{{HO}^{*}}=3.2\pm 0.2\;eV$$
(14)$$\eta_{OER}=\frac{1}{e}max[\left({\Delta G}_{O^{*}}^{o}-{\Delta G}_{{HO}^{*}}^{o}\right),(3.2-\left({\Delta G}_{O^{*}}^{o}-{\Delta G}_{{HO}^{*}}^{o}\right))]-1.23$$

A functional relationship between the *η**_OER_* overpotential and (Δ*G*_*O**_ − Δ*G*_*HO**_) is established to plot the volcanic relationship diagram of the OER, as shown in Figure 4c. The theoretical minimum overpotential of the OER is 0.37 V according to Equation (14).

However, the linear relationship will inherently limit the reaction kinetics of the OER. Therefore, a new catalyst should be designed to circumvent the scaling relations of adsorption energy, allowing it to potentially break the theoretical overpotential cap and exhibit high activity. The strategy for breaking the linear relationship of adsorption energy can be summarized as follows: (1) introducing p-state; (2) introducing a second adsorption site; (3) introducing a proton acceptor group; (4) strain effect; (5) changing the solvent composition; (6) nanoscopic confinement and (7) O–O direct coupling in the absence of HOO* [43]. The different components of the heterojunction can introduce new adsorption sites, so the rational design of the heterojunction can not only improve the performance of the catalyst, but also break the linear relationship of the adsorption energy, leading to excellent catalytic performance.

## 3. Catalysts for Alkaline the HER

### 3.1. Pioneering Design

Nano metal Pt exhibits extremely excellent electrocatalytic HER activity in the acid electrolyte of a proton exchange membrane (PEM) [44]. Unfortunately, Pt is rather rare and PEM is expensive. Therefore, electrolysis of alkaline water is gaining prominence. Pt remains the best catalyst for the alkaline HER, but the HER activity of Pt is orders of magnitude lower at pH = 13 than at pH = 1. The limitations in the Pt HER catalysis arise from the fact that Pt is generally inefficient in the water dissociation step, although it is a good catalyst for the subsequent adsorption and recombination of the reactive hydrogen intermediates (H*). In the research on alkaline water splitting, it was found that the cheap transitional metal oxides were effective for cleaving the HO–H bond, although they were poor at converting H* to H_2_ [45]. Obviously, optimal design of the HER catalysts in the electrolysis of alkaline water can be achieved through the synergetic effects of metals and metal oxides. Two layers of Ni(OH)_2_ clusters with a surface coverage of 35% were electrodeposited preferentially on Pt islands deposited on the Pt(111) electrode, showing eight times higher activity than the pristine Pt(111) electrode. Even without depositing Pt islands, the Ni(OH)_2_/Pt(111) electrode, despite having 35% fewer exposed Pt sites, was also about seven times as active relative to Pt(111) [26]. As expected, Ni(OH)_2_ promoted the water dissociation and thereby enhanced the rate of formation of H* on the nearby vacant Pt surface. Water was firstly adsorbed on the Ni(OH)_2_ sites and dissociated into H* and OH^−^, then the H* was adsorbed on the nearby vacant Pt, corresponding to Equation (2). Finally, two H* species on the Pt surface combined to generate H_2_ and OH^−^ desorbed from the Ni(OH)_2_ domains, corresponding to Equation (3). As Li^+^ interacted more strongly with H_2_O and OH* than K^+^ in alkaline environments, Li^+^ was introduced to probe the HER effect. It was found that the HER on the Ni(OH)_2_/Pt islands/Pt(111) was further enhanced by a factor of 10 in the presence of Li^+^, compared with the pristine Pt(111). However, Li^+^ had no effect on the HER on Pt(111). These results suggested that Ni(OH)_2_ in the presence of Li^+^ played a dual role: one was assisting water dissociation, the other was providing an anchor to hold the beneficial Li^+^ ions in the compact portion of the double layer to form the Ni(OH)_2_–Li^+^–OH–H complexes.

The revolutionary research has disclosed a design, based on the ensemble effect, involving the modification of a metal, a traditional acidic HER catalyst, with an oxyphilic component that can promote the adsorption and decomposition of H_2_O. The H* generated by the oxyphilic component can be adsorbed onto nearby acidic HER active sites and combine into H_2_. Both metal and oxyphilic components catalyze the different intermediate reaction courses, exerting the synergetic ensemble effect.

Subsequent research compared the influence of different metal or metal oxides on the HER in the alkaline electrolytes. It was revealed that Ir exhibited higher HER activity than Pt in the alkaline electrolyte due to the higher rate of the Volmer step, enhanced by more oxyphilic Ir. The difference could be compensated for by the introduction of Ni(OH)_2_, to a certain degree. Ni as a non-noble metal also showed good enough HER activity, which was merely worse than Ir, Pt and Ru but much better than Au [46]. Ni(OH)_2_ still exhibited the highest activity among the transitional metal oxides, including Co(OH)_2_ [47]. Furthermore, it was found that when combined with Pt, β-Ni(OH)_2_ demonstrated better HER activity than α-Ni(OH)_2_ [48].

### 3.2. Metal/Compound Catalysts

The conventional method to prepare the metal/compound catalysts in nanoscale involves reducing the metal ions via impregnation into the metal compound matrix. A typical example was the reduction of the impregnated Pt^2+^ in the pre-synthesized CoS_2_ loaded on a carbon cloth substrate, which delivered a very small HER overpotential of 24 mV at −10 mA cm^−2^ [49]. Another example was the nitridation of the Ru^3+^ pre-impregnated Ni(OH)_2_ on Ni foams, giving rise to the product of Ru/Ni_3_N [50], suggesting that noble metals are hard to be nitridated. If there are no substrates, such as carbon clothes and Ni foams, surfactants could be used to separate metal nanoparticles. Oleylamine is not only a surfactant but also a reducing agent. A core-shell structure of Pd@Fe_3_O_4_ was produced with oleylamine by a two-step reaction, and the shell was further oxidized to FeO_x_(OH)_2−2x_ [51].

Another common method to prepare metal/compound catalysts is electrodeposition. Ru_1_/NiFeAl LDH (layered double hydroxides), where Ru_1_ represents Ru single atoms, could be supported on Ni foams by electrodeposition with lower concentrations of Ru^3+^ and Al^3+^ salts. After immersing Ru_1_/NiFeAl LDH in concentrated NaOH, Al in Ru_1_/NiFeAl LDH was etched, resulting in Ru_1_ on defective NiFe LDH (Figure 5) [52]. Only 1.2 wt% of Ru single atoms on defective NiFe LDH delivered an HER overpotential of 18 mV at −10 mA cm^−2^, which was lower than those of the control samples of Ru_1_/NiFe LDH and defective NiFe LDH. The very high HER activity of the Ru_1_ on defective NiFe LDH could arise from the ensemble effect of metal and compound, defects in the compound and surface effects from single atoms. Instead of etching Al to produce defects, hydroxylation was found to improve the HER kinetics. NiOOH was firstly electrodeposited on the Cu mesh substrate. The NiOOH on Cu was annealed in Ar to form new NiO/Cu_2_O/Cu heterointerfaces. These new interfaces were finally converted to (HO)_x_-NiO/Cu heterojunctions by repeated cyclic voltammetry (CV) [53]. Despite the complex craft, (HO)_x_-NiO/Cu exhibited a small HER overpotential of 33 mV at −10 mA cm^−2^, suggesting that chemical hydroxylation and the heterointerface exerted a synergetic effect on the improvement of HER activity.

An ion-exchange method was applied to exchange the active metal ions into a desirable compound with a special structure. Hydroxyapatite (Ca_10_(PO_4_)_6_(OH)_2_, HAP) is an abundant inorganic phosphorous mineral. HAP nanowires had a high aspect ratio which endowed HAP with excellent membrane-forming ability. The active metal precursor of Ru^3+^ was introduced into the HAP skeleton by exchanging it with Ca^2+^. During annealing in H_2_/Ar, Ru and Ru_2_P were produced in order. In situ XRD revealed that the (100) of Ru was gradually changed to (112) of Ru_2_P. The Ru/Ru_2_P had an HER overpotential of only 24 mV at −10 mA cm^−2^_,_ with a very small Tafel slope of 31.99 mV dec^−1^ [41]. In contrast, the control samples of Ru and RuP merely possessed an overpotential of 55 and 116 mV with Tafel slopes of 93.02 and 131.17 mV dec^−1^.

Since metal Ni or Co was also active for the HER in the alkaline electrolyte, an interesting method of in situ exsolution of Ni was applied in the heterojunctions for the HER. An Ar low-pressure annealing of Ni(OH)_2_/carbon nanotubes (CNTs) prepared via low-temperature hydrolysis resulted in a partially in situ reduced Ni core and NiO shell [54]. Impressively, the NiO/Ni/CNT catalyst achieved an HER overpotential below 100 mV at −10 mA cm^−2^, evidently smaller than that for Ni/CNT and much smaller than for NiO/CNT. Similarly, in situ exsolution of Ni also occurred via the annealing of Ni-doped CeO_2_ under an H_2_/Ar atmosphere [55]. A more facile method to segregate Ni is the annealing of the Ni^2+^/Ce^3+^ nanofibers via electrospinning [56]. When using elements which are prone to forming carbides, segregated Ni and carbides were produced on carbon-based substrates. In the case of hydrothermally obtained NiMoO_4_ nanorods as precursors coated by polydopamine, the size of Ni particles in Ni/Mo_2_C/C was beyond nanoscale [57]. In the solid phase reaction involving (NH_4_)_6_Mo_7_O_24_·4H_2_O, Co(CH_3_COO)_2_·6H_2_O and melamine, β-Mo_2_C nanoparticles (ca. 18 nm) surrounded by Co nanoparticles (ca. 7 nm) were encapsulated in the produced tubular nanocarbon [58]. The Ni-substituted polyoxometalate, [Ni(en)_2_(H_2_O)_2_]_6_[Ni_6_(Tris)(en)_3_(BTC)_1.5_(*B*-α-PW_9_O_34_)]_8_·12en·54H_2_O, was prepared on reduced graphite oxide (GO). After the annealing of the NiW compound in N_2_, the Ni element was segregated in situ to become metal, while the W element was reduced to WC. Meanwhile, reduced graphite oxide was doped by N [59]. A representative Ni/WC hybrid particle could be seen, suggesting a distinct contact between Ni and WC to form the heterointerfaces. Based on the good lattice match between two rotated (120) planes of Co and one (120) plane of Ni_3_N, a deliberated design of the heterogeneous junction between Co and Ni_3_N was executed via an in situ exsolution of Co from NiCo_2_O_4_ under an NH_3_ atmosphere [60]. The TEM image revealed another good lattice match between (100) planes of Co and the (002) plane of Ni_3_N.

The partial conversion of nano-metals to metal compounds could also form Schottky heterojunctions. The noble metal Ru was generated via the reduction of Ru^3+^ with NaBH_4_. Subsequent calcination at low temperature partially converted Ru to RuO_2_ [61]. The Ru/RuO_2_ exhibited an HER overpotential of 34 mV at −10 mA cm^−2^, much smaller than the 139 mV for pure Ru. In the case of a non-noble metal Co^2+^ as a precursor, Co was formed during the heating treatment of the obtained non-metal state of Co and C precursor. The subsequent phosphorylation made partial Co be converted to CoP [62]. Another example was the nitridation of electrodeposited porous Ni microspheres on Ni foams to form the Ni_3_N/Ni heterojunctions [63]. The optimal control of temperature and time could generate the Ni_3_N phase with an increased yield due to the low thermal stability of Ni_3_N beyond 350 °C (Figure 6a–e). The linear sweep voltammetry (LSV) experiments were conducted at 5 mV s^−1^ using a three-electrode configuration. All LSV curves were *IR* corrected. The monolithic Ni_3_N/Ni on Ni foams was directly used as the working electrode. A calibrated Ag/AgCl (saturated KCl) with salt bridge kit and a carbon rod were used as the reference and counter electrodes, respectively. The electrolyte for the HER was 1.0 mol L^−1^ KOH, which were bubbled with H_2_ throughout the whole electrochemical experiments. Finally, the Ni_3_N/Ni on Ni foams possessed a very low HER overpotential of only 12 mV to deliver a current density of −10 mA cm^−2^ in KOH, which was lower than those for Ni and Pt/C on Ni foams, as shown in Figure 6f.

### 3.3. Compound/Compound Catalysts

Although most of the metal/compound catalysts mainly rely on the ensemble effect, heterojunctions, if evident, may also be effective in improving the HER activity of the catalysts to a certain extent. In comparison, in compound/compound catalysts, the interface-induced electronic effect dominates the promotion of HER activity. Molybdenum sulfides are earth-abundant HER electrocatalysts with high catalytic activity and low cost [64]. In particular, the edges of MoS_2_ were identified as active sites for the HER [65]. The sodiation/desodiation treatment could generate more unsaturated sulfur edges, which was demonstrated to be able to improve the HER activity of MoS_2_ [66], although the process was complex and costly. MoS_2_ on carbon clothes was prepared by hydrothermal sulphuration treatment with phosphomolybdic acid as a precursor. The controllable phosphorylation of the MoS_2_ was realized by annealing with NaH_2_PO_2_ to form the MoP/MoS_2_ heterojunctions [67]. The HER overpotential of the MoP/MoS_2_ achieved 54 mV at −10 mA cm^−2^. Similar processes, but with treatment at lower temperatures using Ni foams as the substrates, resulted in the formation of amorphous flower-like MoP/MoS_2_ [68]. The hydrophobic/aerophilic nature of the Ni foam surface was thoroughly changed to a hydrophilic/aerophobic nature, which was beneficial for strong water absorption and rapid gas evolutions. Hydrothermal sulphuration and subsequent nitridation from polypyrrole-coated MoO_3_ produced Mo_5_N_6_-MoS_2_ on the hollow carbon nanoribbons, which showed a small HER overpotential of 53 mV at −10 mA cm^−2^ [69]. Sulphuration of MoO_3_ and another different metal salt precursor of Ni(OH)_2_ on carbon clothes by hydrothermal treatment yielded amorphous 1T-MoS_2_/crystalline Ni_3_S_4_ interfaces. Note that the generated nickel sulfide was Ni_3_S_4_ rather than Ni_3_S_2_ [70]. The MoS_2_/Ni_3_S_4_ delivered a small HER overpotential of 44 mV at −10 mA cm^−2^, suggesting that Ni_3_S_4_ was also supposed to be a good catalyst for the HER.

Mo_2_C can be used as a good HER catalyst [71]. The pompon-like Mo_2_C particles were obtained via calcination of the dried mixture of Na_2_MoO_4_ and dopamine as a carbon source at 800 °C for 2 h in Ar. The conventional heating treatment of Mo_2_C in NH_3_ was supposed to produce multiple lattice-mismatched Mo_2_C/Mo_2_N heterostructures. Nevertheless, it was surprising that the well lattice-matched Mo_2_C/Mo_2_N heterointerfaces were prepared via a gradient heating treatment of the Mo_2_C particles at 650 °C for 1 h and 800 °C for 2 h in NH_3_ atmosphere, as shown in Figure 7a [28]. However, the height difference between Mo_2_C and Mo_2_N detected using an atomic force microscope identified the obvious heterointerface. The Kelvin probe force microscopy revealed the potential distribution of Mo_2_C/Mo_2_N heterostructures (Figure 7b), suggesting that the electrons tended to transfer from Mo_2_C to Mo_2_N in alkaline environments, which is a great achievement. The lattice-matched Mo_2_C/Mo_2_N heterostructures realized an HER overpotential of 80 mV at −10 mA cm^−2^, which was much better than 147 mV for lattice-mismatched Mo_2_C/Mo_2_N, 258 mV for Mo_2_C/Mo_2_N and 420 mV for Mo_2_C/Mo_2_N. Another Mo_2_C/Mo_2_N heterostructure was obtained by utilizing carbon clothes as the substrate and carbon source without mixing other carbon-rich substances such as dopamine [72]. The Mo_2_C/Mo_2_N on the carbon clothes showed a lower HER overpotential of 54 mV at the same current density.

Cobalt phosphides are also popular HER catalysts [73]. A precursor consisting of Co and P was prepared via a hydrothermal method with an organophosphorus coupling molecule. The reduction of the precursor with H_2_ at different temperatures gave rise to different products. At 800 °C, the complete reduction of cobalt phosphonate was finished, resulting in CoP/Co_2_P interfaces on N,P-doped carbon. When the reduction temperature increased to 900 °C, the product turned to be the pure Co_2_P [74]. The CoP/Co_2_P on the doped carbon had an HER overpotential of 109 mV at −10 mA cm^−2^, which was lower than 183 mV for the pure CoP on the carbon. The construction of interfaces between different cobalt phosphide phases depended on the careful optimization of craft parameters. The more conventional method is the phosphorylation of a dual-metal precursor, such as NiCo LDH. The Co_2_P-Ni_2_P delivered an HER overpotential of 90 mV at −10 mA cm^−2^ [75].

MXenes, typically Ti_3_C_2_ [76,77], as 2D laminated materials, are always used as the supports for catalysts. The phosphorylation of Co LDH on the Ti_3_C_2_ MXene support produced the CoP/MXene catalyst. Although the MXene had poor HER activity, the CoP with 40% mass loading on MXene exhibited an HER overpotential of 102 mV to drive a current density of −10 mA cm^−2^ [78]. If the MXene and Co LDH were coated on Ni foams by electrophoretic deposition in order, phosphorylation led to the generation of the Co_2_P/MXene heterojunctions [79]. The HER overpotential of the Co_2_P/MXene on the Ni foams was only 42 mV at −10 mA cm^−2^.

### 3.4. The Performance of Catalysts

The important properties of the HER catalysts should include overpotentials at low and high current densities, Tafel slopes and stability. It seemed that the cyclic performances of most catalysts used in labs over relatively short times were good enough. Hence the overpotentials and Tafel slopes are very crucial for HER catalysts. Table 1 lists the overpotentials and Tafel slopes of different catalysts.

## 4. Catalysts for Alkaline the OER

### 4.1. Ir(Ru)-Based Composite Catalysts

Compared with the HER, the OER kinetics are more sluggish, meaning that the OER overpotential dominates the extra voltage of overall water electrolysis, even in acidic electrolytes. As mentioned before, the theoretical minimum overpotential of the OER is 370 mV, according to the scaling relation. The benchmark catalysts of the OER are RuO_2_ and IrO_2_, which can achieve overpotentials less than 750 mV in acidic electrolytes at 5 mA cm^−2^ [97]. Based on the consideration that an Ir-based perovskite material of SrIrO_3_ was attempted as an OER catalyst instead of IrO_2_, an epitaxial SrIrO_3_ thin film was deposited on SrTiO_3_ substrates via pulsed laser deposition [15]. During the alternate CV and galvanostatic methods, lasting for 2 h in the acidic electrolyte, the overpotentials of the film continuously changed from 340 to only 270 to 290 mV at 10 mA cm^−2^. The value after testing not only outperformed the control samples of the IrO_2_(110) film on TiO_2_(110) and IrO_2_ nanoparticles, but also significantly exceeded the theoretical 370 mV overpotential of the OER. It was found that above 30% of the Sr was leached into the electrolyte to form rough IrO_3_ or anatase IrO_2_ after testing, as confirmed by AFM and XPS. The theoretical calculation also demonstrated that three monolayers of IrO_3_ on SrIrO_3_ had an overpotential lying at the volcano peak, which was consistent with the improved OER activity after leaching the surface Sr in the SrIrO_3_.

Although the IrO_x_/SrIrO_3_ film was investigated in an acidic electrolyte, the Ir-based composites could also be effective for the OER in alkaline electrolytes, which was proven by the subsequent research on Ir/Co(OH)_2_ prepared via the reaction of the two precursors with NaBH_4_ [98]. Two types of Ir species, including Ir single atoms and Ir clusters, were anchored on Co(OH)_2_, showing the coexistence of Ir^0^ and Ir^4+^. The K_2_IrCl_6_ could be also reduced with glycol, resulting in uniform Ir nanoparticles with an average size of 1.5 nm deposited on the ex situ-produced Ni(OH)_2_ [30]. During the OER, Ni(OH)_2_ was converted to NiOOH, while Ir nanoparticles on the surface were oxidized to IrO_x_. In the Ir-based composite, the adsorptions of the OER intermediates were no longer restricted to the active sites of the single component, and hence the OH* and O* species were adsorbed on the active sites of Ni(OH)_2_/NiOOH and IrO_x_, respectively. The combination of the two intermediates at the heterostructure interface accelerated the formation of OOH*, which eventually broke the scaling relation in the OER kinetics. Finally, O_2_ was generated via the desorption of OOH*. Therefore, the Ir/Ni(OH)_2_ required overpotentials of only 224 and 270 mV at 10 and 100 mA cm^−2^, respectively. In contrast, the overpotentials were 437 mV for Ni(OH)_2_ at 10 mA cm^−2^ and 343 mV for Ir nanoparticles at 100 mA cm^−2^. The calcination of chloroiridic acid impregnated NiO on carbon clothes via immersion generated a high loading (18 wt%) of Ir single atoms (Ir_1_) on the surface of NiO on carbon clothes (Figure 8a–c) [99]. Accordingly, the majority of Ir atoms on the NiO support were around the 4+ oxidation state, indicating that Ir single atoms were highly oxidized. The Ir_1_/NiO on carbon clothes delivered 215 mV at 10 mA cm^−2^ with a Tafel slope of 38 mV dec^−1^, while IrO2 and NiO required 294 mV with 110 mV dec^−1^ and 370 mV with 113 mV dec^−1^. The DFT calculation demonstrated that the rate-determining step of the OER changed from the oxidation of OH* to O* on the NiO(001) with 0.98 eV of Δ*G* to the transformation of O* to OOH* on Ir/NiO(001) with 0.67 eV of Δ*G*, as shown in Figure 8d. The theoretical calculation [99] was also highly consistent with the explanation for Ir/Ni(OH)_2_ [30]. It was expected that NiFeP replacing NiO or Ni(OH)_2_ could further improve the OER activity. Nano porous (Ni_0.74_Fe_0.26_)_3_P was synthesized by electrochemically selective etching of the corresponding alloy ingot. Atomic iridium was electrodeposited on the nano-porous (Ni_0.74_Fe_0.26_)_3_P via 750 CV cycles. It was interesting that additional 50 CV cycles in fresh KOH (1 mol L^−1^) lead to a surface self-reconstruction, resulting in the partial conversion of (Ni_0.74_Fe_0.26_)_3_P to Ni(Fe) (oxy)hydroxides (NiFeO in short) [100]. Consequently, the overpotential of the Ir/NiFeO was as low as 197 mV at 10 mA cm^−2^.

The calcination of impregnated Ru^3+^ in Ni(OH)_2_ on Ni foams at 300 °C produced the RuO_2_/NiO on Ni foams [101]. The RuO_2_/NiO on Ni foams exhibited an overpotential of 250 mV at 10 mA cm^−2^, which was 33 mV smaller than that of RuO_2_ on Ni foams. The calcination of the product resulting from the reaction between Ru^3+^, Co^2+^ and NaBH_4_ at 600 °C yielded the RuO_2_/Co_3_O_4_ catalysts with O vacancies, which possessed an optimal overpotential of 152 mV at 10 mA cm^−2^ [102]. In contrast to inorganic RuO_2_, an interesting organics/inorganics composite was also used as the OER catalysts. The Ru-rich conductive RuNiCo MOF as branches were grafted onto the Co-rich CoNiRu LDH as tree trunks by a ligand of 2,3,6,7,10,11-hexaiminotriphenylene (HITP) with rich amino groups to assemble the CoNiRu-based nanotrees, as shown in Figure 9 [103]. Owing to the specific tree structure facilitating the access of the electrolyte ions, the overpotentials of CoNiRu-based nanotrees were only 255 and 335 mV to reach the active current densities of 20 and 100 mA cm^−2^, respectively.

### 4.2. NiFe-Based Composite Catalysts

As we know, NiFe LDH is one of the most cost-effective catalysts for the OER in alkaline electrolytes [104]. Graphene is a 2D monolayer of graphite, providing unique mechanical properties, high conductivity and a large theoretical specific surface area [105,106,107,108,109]. The strong coupling of NiFe LDH and graphene layer by layer was expected to fully utilize the specific surface area of each component. However, the carbonate ions in the NiFe-CO_3_ LDH nanoplates prepared via a simple hydrothermal method exhibit a rather high affinity to the hydroxide layers, making delamination quite challenging. The conversion of FeNi-CO_3_ LDH into NiFe-Cl LDH could be achieved via ion exchange, facilitating the access of GO between LDH layers. The maximal exfoliation of LDHs and assembly with GO were realized by ion exchange over ten days. Subsequently, the NiFe-rGO LDH was produced by reducing the assembled NiFe-GO LDH with N_2_H_4_‧H_2_O, demonstrating an OER overpotential of 206 mV at 10 mA cm^−2^ [110]. More popularly, Ni(OH)_2_ or NiO could be deposited on the heteroatom-doped graphene. It was found that the trace of Fe etched from the beaker could increase the CV peaks and OER currents. Hence, an optimized 12 μmol L^−1^ FeCl_3_ was deliberately added to the KOH electrolyte, causing the Fe that was bonded at the neighboring sites of Ni centers on the doped graphene [111].

Besides graphene, metals could also be constructed in NiFe-based catalysts. A uniform electromagnetic field as an external field was used to assist alignment growth of the Ni_x_Fe_1-x_ nanowires on the nickel foam substrate. Then ultrathin NiFe oxyhydroxide layers were formed in situ on the surface of the NiFe alloy nanowires (Figure 10) [112]. The LSV experiments were carried out at 5 mV s^−1^ using a three-electrode configuration. Chronopotentiometry measurements were executed to evaluate the long-term stability. The nanowires on Ni foams were used as the working electrode. A calibrated Ag/AgCl (3 mol L^−1^ KCl) and a graphite rod were used as the reference and counter electrodes, respectively. The electrolyte for the OER was 1 mol L^−1^ KOH, which was bubbled with H_2_. The IR compensation was performed using the automatic current interrupt method via the CH instrument 660E working station. For the OER, in order to avoid overlap between Ni^2+^/Ni^3+^ oxidation and the OER, polarization curves were recorded from high initial potentials to low final potentials with 5 mV s^−1^ scan rate. The NiFe@NiFeOOH heterostructure delivered an OER overpotential of 190 mV at 10 mA cm^−2^. Surprisingly, the NiFe@NiFeOOH heterojunctions yielded stably the overpotentials of only 248 and 258 mV over 120 h at current densities of 500 and 1000 mA cm^−2^, respectively, which was the best report.

A new method in which a pulsed laser ablated the NiFe alloy target in the urea solution was used to prepare the Ni/NiFe LDH precursor. Electrochemical oxidation of the Ni/NiFe LDH precursor resulted in the NiO/NiFe LDH heterostructure [113]. The NiO/NiFe LDH on Cu foam achieved an OER overpotential of 205 mV at 30 mA cm^−2^. In the NiO/NiFe LDH heterostructure, a special type of nickel cations, denoted by S1 in the inset of Figure 11a, located at the NiO/NiFe LDH intersection and neighboring the lattice O of LDH atop the Fe–Ni–Ni triangle, displayed a smaller OER overpotential in the volcano curve (Figure 11a). The breakthrough of scaling relation was due to the O* bridged to NiO and neighboring NiFe LDH simultaneously, as shown in Figure 11b.

An interesting example of the NiFe LDH preparation was the spontaneous growth on FeS nanosheets covering iron foams [114]. The Fe–Fe bond in FeS was deemed to be very active and could be easily oxidized in the solution with oxygen, which resulted in the spontaneous growth. The NiFe(OH)_x_/FeS on iron foams exhibited an OER overpotential of 245 mV at 50 mA cm^−2^, which was superior to 295 mV for NiFe(OH)_x_ and 339 mV for FeS on iron foams. It was also better than the 362 mV for IrO_2_ on iron foams at the same current density.

### 4.3. Other Compound/Compound Catalysts

Ni_3_S_2_, especially Fe-doped Ni_3_S_2_, is also a good OER catalyst [115,116]. A representative example of compound/compound heterostructure is Co_9_S_8_/Ni_3_S_2_ nanotubes, which was formed through the sulfurization of Co(OH)(CO_3_)_0.5_ nanoneedles growing on Ni foams using a hydrothermal method. The Co_9_S_8_/Ni_3_S_2_ nanotubes delivered an OER overpotential of 281 mV at 50 mA cm^−2^ [117].

The nitration of cobalt oxide hydrothermally grown on carbon papers in Ar yielded CoN. Then Co(OH)_2_ was electrodeposited on the CoN electrode. Abundant oxygen vacancies were generated by vacuum desiccation of the Co(OH)_2_/CoN on carbon papers [118]. The Co(OH)_2_/CoN electrocatalyst with oxygen vacancies exhibited an OER overpotential of 206 mV at 10 mA cm^−2^, significantly lower than 237 mV for intact Co(OH)_2_/CoN. It was no doubt that the free energy of the reaction coordinate with Co(OH)_2_/CoN in the rate-dependent step was smaller than those of the individual components, as shown in Figure 12a. Unfortunately, the free energies of the reaction coordinate with both the oxygen vacancy and Co sites in Co(OH)_2_/CoN in the rate-dependent step were larger than that with the intact Co(OH)_2_/CoN. It was stated that the OH* intermediate migrated from the reaction path of the oxygen vacancy to the Co sites in Co(OH)_2_/CoN through a relatively low barrier of 1.57 eV, compared with 1.89 eV with the intact Co(OH)_2_/CoN (Figure 12b,c).

Metal Organic Frameworks (MOFs) are crystalline porous solids composed of metal ions held in place by multidentate organic molecules [119]. Poly-[Co_2_(benzimidazole)_4_] (PCB) is a laminar MOF comprising CoN_4_ structures. In fact, layered MOFs were challenging to exfoliate via ultrasonication, despite the weak coordination bonds in each layer [120]. Without exception, the PCB microcrystals could not be exfoliated. When adding Fe^3+^ ions in the PCB precursors, CoFeO_x_ nanoparticle-embedded PCB was obtained, which could be exfoliated to form monolayered nanosheets by ultrasonication with a yield of 25% [121]. The overpotential of the exfoliated nanoparticle-embedded PCB monolayers decreased to 232 mV from 316 mV for nanoparticle-embedded PCB at 10 mA cm^−2^.

### 4.4. Bifunctional Catalysts and OER Performance

As most noble metal catalysts had relatively low HER activities in alkaline electrolytes, the incorporation of another component to form the heterostructures was required to enhance the HER catalytic performance. However, compared with the HER overpotential, the OER overpotential plays a dominant role in determining the overall voltage of water electrolysis. Hence it is feasible that the incorporated component could be an OER component instead of the HER one. Considering the preparation cost of catalysts, bifunctional catalysts which comprise the HER active components and the OER ones were desirable in the alkaline electrolysis of water. The typical bifunctional catalysts are MoS_2_/Ni_3_S_2_ [122,123], MoS_2_/Co_3_O_4_ [124] and CoP/Co_3_O_4_ [125].

Similar to the HER, the overpotentials and Tafel slopes are very crucial parameters for the OER and bifunctional catalysts. For certain catalysts, where the redox peaks were beyond a benchmark current density of 10 mA cm^−2^, the OER overpotentials were also compared at larger current densities. Table 2 and Table 3 list the catalytical performances of different catalysts.

## 5. Conclusions and Perspective

Heterojunction engineering has been one of the most promising strategies to develop efficient electrocatalysts for alkaline water electrolysis. The composite catalysts with two-phase heterojunctions exhibit both the ensemble effect and electronic effect, which accelerate the kinetics of the HER or/and OER. The ensemble effect utilizes the multi-sites of different components to accelerate the different intermedium reactions. In the electronic effect, the d-band center theory was proposed to be associated with the adsorption and desorption energies of intermediates and catalyst. Besides, from the perspective of work function, the modulation of electron density at heterogeneous junctions promotes various catalytic reactions. For the more multi-electron OER, a scaling relation that the free energy difference between HOO* and HO* is 3.2 eV is built. The theoretical minimum overpotential of the OER is 0.37 V.

The Ni(OH)_2_/Pt catalyst overcomes the drawback of the inefficiency of pure Pt in the water dissociation step in the alkaline HER by incorporating the Ni(OH)_2_ active component. Consequently, the synergistic effect in the Ni(OH)_2_/Pt catalyst significantly improves the HER activity, revealing the importance of heterojunction engineering. Other metals, including non-noble metals, combined with different compounds, constituted the catalysts with heterojunctions. In particular, single atomic Ru on defective NiFe LDH showed only 18 mV of HER overpotential at −10 mA cm^−2^. The ion-exchange and in situ exsolution were new methods to prepare the composite catalyst made of metal and the corresponding compound. Certainly, other HER active compounds, such as MoS_2_, can replace metal to produce compound/compound catalysts, which could lower the cost.

Atomic Ir specimens, including single atomic Ir, were combined with compounds to enhance OER efficiency. During the OER, the surface of Ir was oxidized to IrO_x_. Different intermediate species, such as OH* and O*, in the OER were adsorbed on various active sites within the Ir-compound catalysts. The combination of these two intermediates at the heterointerface accelerated the formation of OOH*, ultimately breaking the scaling relation in the OER kinetics. A tree structure composed of the rich Ru conductive RuNiCo MOF as branches and the rich-Co CoNiRu LDH as tree trunks present an interesting design to further increase the specific surface. NiFe LDH and Ni_3_S_2_ as the representative of non-noble metal catalysts for the OER were used to form the heterojunction with other compounds. As an interesting example, NiFe LDH spontaneously grew on the FeS nanosheets. Considering the cost and advantage of the catalysts with heterojunctions, the encouragement of bifunctional catalysts comprising both an HER component and an OER one is recommended.

The development trend in alkaline water electrolysis involves reducing the amount of noble metal or utilizing a non-noble metal or even its compounds in heterojunction engineering. Atomic Pt and Ir, even single atom ones, remain the most efficient catalysts for the HER and OER, respectively, contributing to cost reduction by minimizing the use of noble metals. Ni and MoS_2_ show promise as alternatives to Pt in two-phase catalysts for the HER, while NiFe LDH and Fe-doped Ni_3_S_2_ are considered promising alternatives to Ir for the OER. The preparation of a monolayer or a few layers of the laminated materials is worth studying. However, the compromise between performance and cost should be considered.

The Ni foams are very good substrates to provide Ni sources without binders. Hydrothermal treatment is a facile and efficient method for growing catalysts on Ni foams in situ. The partial conversion of the grown catalysts under precise control are promising for forming heterogeneous junctions. If it is not necessary to utilize Ni foams to provide Ni sources, impregnation, ion-exchange and in situ exsolution are also future directions for preparing the heterojunctions. Additionally, multi-metal co-doping could be applied to promote the electrocatalytic performance of heterojunctions [140].

From an industrial perspective, the much longer stability of these nano-catalysts with heterojunctions needs to be measured. The binding capability of the composite catalysts and substrates should be regarded as a scientific problem in long term electrolysis. The two-phase materials with more facile craft, such as 3D printing, may be exploited. Additionally, these composite catalysts can be also applied in new electrolysis technology, for instance, alternate alkaline water electrolysis [9,141,142].

## Figures and Tables

**Figure 1 materials-17-00199-f001:**
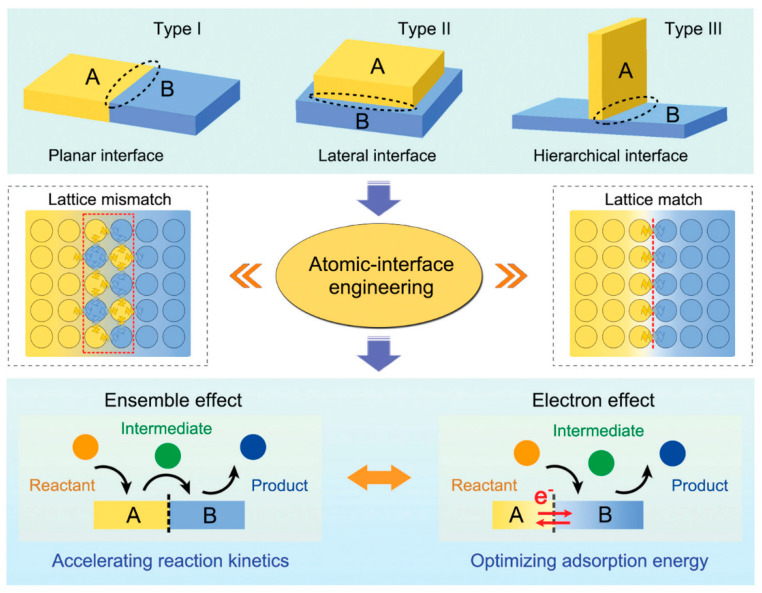
Three heterojunction types and characteristic effects of atomic heterojunction engineering. Reprinted from reference [25], copyright (2021), RSC.

**Figure 2 materials-17-00199-f002:**
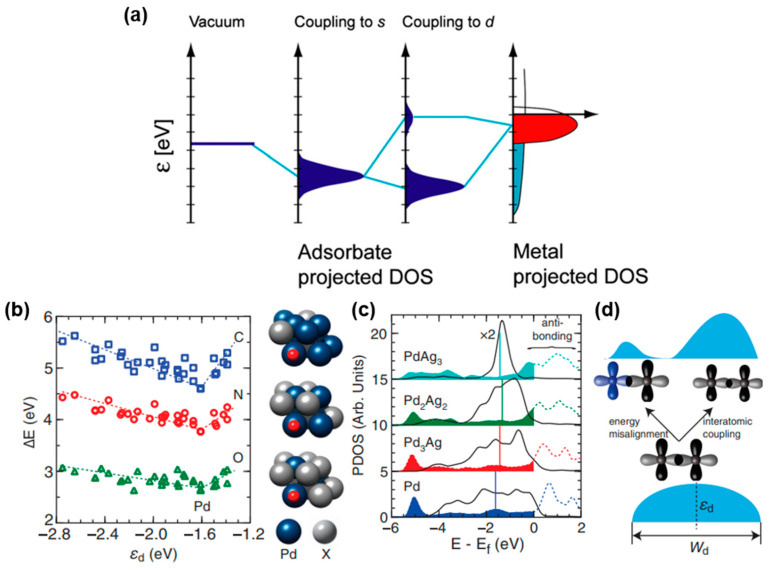
(**a**) Schematic illustration of formation of a chemical bond between adsorbate valence level and s and d states of transition-metal surface. Reprinted from reference [34], copyright (2005), Springer. (**b**) Adsorption energies of C, N and O on (111) surface of Pd and Pd/X alloys. (**c**) Projected DOS onto Pd 4d orbitals (solid lines) of Pd and Pd/Ag alloys and coupled C 2p orbital (dashed lines) of C on Pd and Pd/Ag alloys, vertical solid lines denote d-band center of Pd atom in (111) surface of Pd and Pd alloys. (**d**) Schematic illustration of underlying mechanisms for variations in d-band shape upon formation of alloys. Reprinted from reference [36], copyright (2014), APS.

**Figure 3 materials-17-00199-f003:**
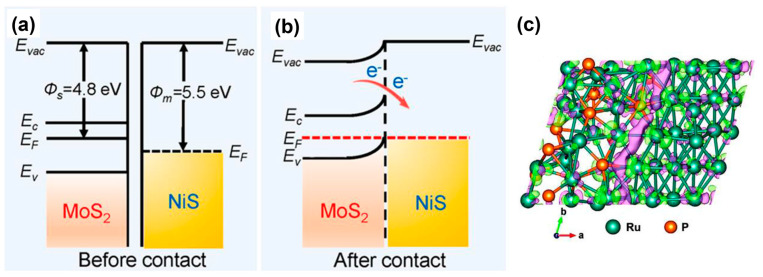
Energy band diagrams of metallic NiS and n-type semiconductor MoS_2_ (**a**) before and (**b**) after formation of a Mott-Schottky heterojunction, where *E*_vac_, *E*_F_, *E*_c_ and *E*_v_ represent vacuum energy, Fermi level, conduction band and valence band, respectively. Reprinted from reference [38], copyright (2022), Elsevier. (**c**) Charge density difference plots at the Ru-Ru_2_P heterointerface. The magenta and green colors represent the charge accumulation and depletion regions, respectively. Reprinted from reference [41], copyright (2023), Wiley.

**Figure 4 materials-17-00199-f004:**
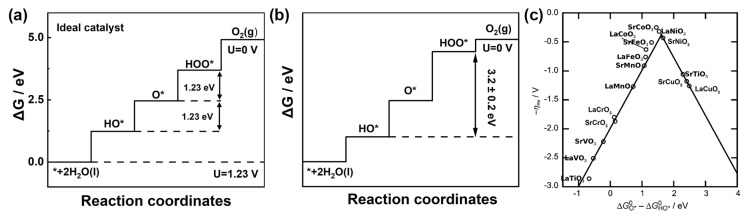
Standard free energy diagram and for the OER at zero potential, an equilibrium potential of 1.23 V and the potential for which all steps become downwards over (**a**) an ideal catalyst and (**b**) an LaMnO_3_ catalyst with a scaling relationship. (**c**) The scaling relationship between the free energy of the water dissociation on top of oxygen (ΔG HOO* − ΔG HO* (eV)) and the free energy of proton removal (ΔG _O*_ − ΔG _HO*_ (eV)). Reprinted from reference [42], copyright (2011), Wiley.

**Figure 5 materials-17-00199-f005:**
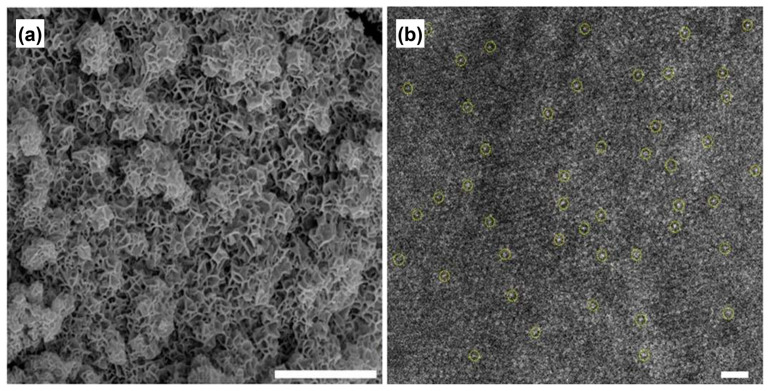
(**a**) SEM and (**b**) aberration-corrected TEM images of Ru_1_/defective NiFe LDH. Single atoms are indicated by yellow circle. Scale bar of (**a**) 1 μm and (**b**) 1 nm. Reprinted from reference [52], copyright (2021), Springer Nature.

**Figure 6 materials-17-00199-f006:**
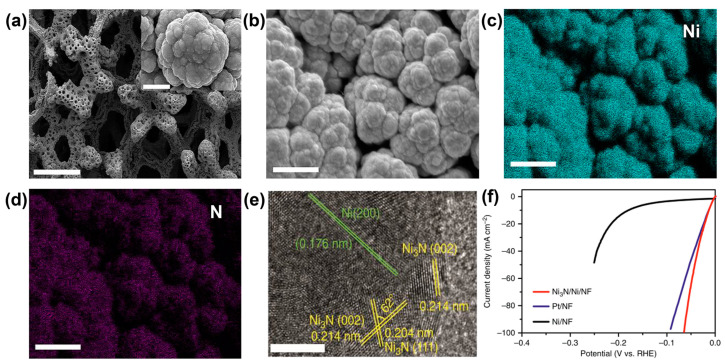
SEM images of Ni_3_N/Ni/NF at (**a**) low and (**b**) high magnifications and elemental mapping images of (**c**) Ni and (**d**) N. (**e**) HRTEM image of the Ni_3_N/Ni interface. Scale bars, (**a**) at 500 μm; inset of (**a**) at 3 μm; (**b**–**d**) at 10 μm; and (**e**) at 5 nm. (**f**) LSV curves of Ni_3_N/Ni/NF, Ni/NF and optimized Pt/NF (Pt/C: 2.5 mg cm^−2^) for the HER in 1.0 mol L^−1^ KOH. Reprinted from reference [63], copyright (2018), Springer Nature.

**Figure 7 materials-17-00199-f007:**
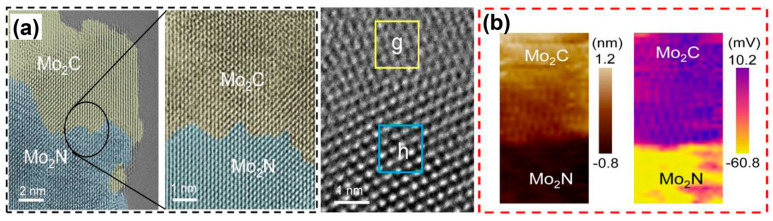
(**a**) Aberration-corrected TEM and (**b**) AFM (**left**) and KPFM (**right**) images of Mo_2_C/Mo_2_N heterojunctions. Frame g in (**a**) is Mo_2_C and Frame h is Mo_2_N. Reprinted from reference [28], copyright (2022), Wiley.

**Figure 8 materials-17-00199-f008:**
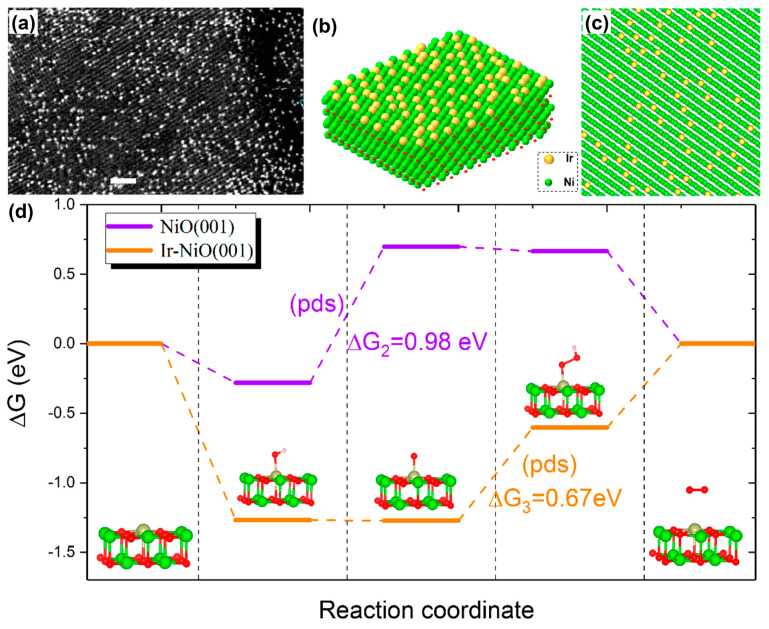
(**a**) Representative HAADF-STEM image of Ir_1_/NiO catalyst and (**b**,**c**) corresponding atomic models. (**d**) Free energy diagrams of the OER at 1.23 V vs. RHE on perfect NiO(001) and single Ir atom-doped NiO(001) (Ir/NiO(001)). Red balls are O atoms. PDS is an abbreviation of potential determining step. Reprinted from reference [99], copyright (2020), ACS.

**Figure 9 materials-17-00199-f009:**
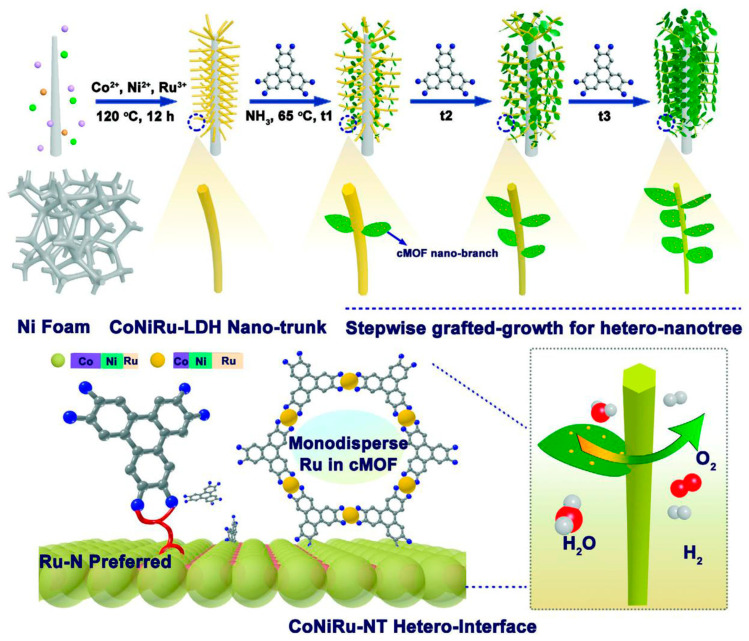
Schematic illustration of the rational stepwise design of CoNiRu-based MOF/LDH hetero-nanotree catalysts for overall water splitting. Reprinted from reference [103], copyright (2022), Wiley.

**Figure 10 materials-17-00199-f010:**
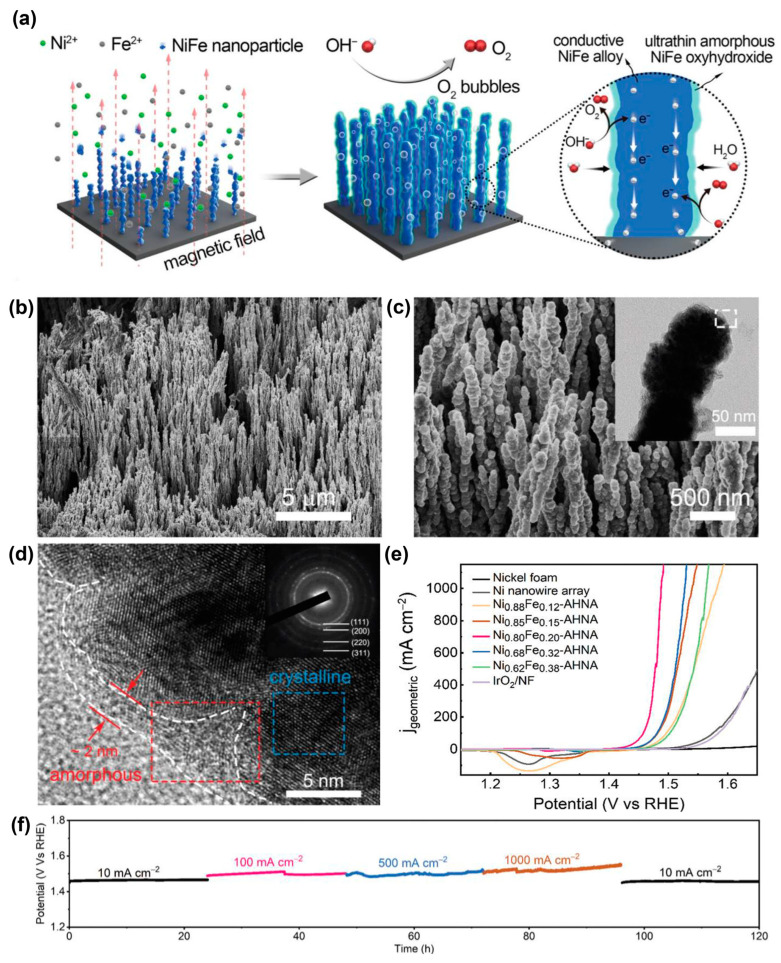
(**a**) Schematic illustration of the synthesis of Ni_x_Fe_1-x_ nanowire arrays. SEM images at (**b**) low and (**c**) high magnifications, inset of (**c**) TEM image of a single nanowire, (**d**) HRTEM image of Ni_x_Fe_1-x_ nanowire arrays, inset of (**d**) SAED pattern, (**e**) polarization curves recorded on different catalysts and (**f**) chronopotentiometric curves of Ni_0.8_Fe_0.2_ nanowire arrays in 1 mol L^−1^ KOH at various current densities. Reprinted from reference [112], copyright (2020), RSC.

**Figure 11 materials-17-00199-f011:**
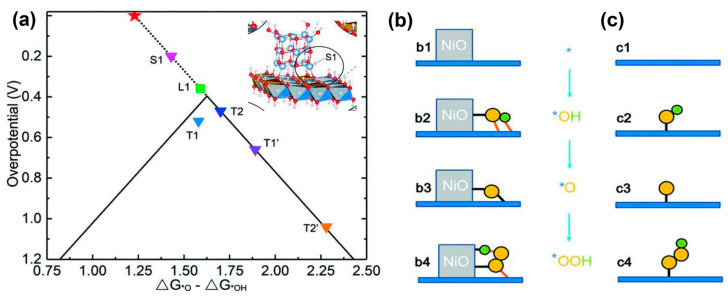
(**a**) Volcano curve of the theoretical overpotential for OER processes. Triangles represent the overpotentials for different NiO/NiFe LDH models and the green square represents the Fe site (L1) at pure LDH edge. Inset of (**a**) is S1 (grey for Ni, brown for Fe, red for O and white for H) and T1, T2, T1′ and T2′ represent Fe site at the edge of LDH with the NiO cluster located nearby and the other edge ions of LDH being passivated by *OH or H_2_O. The red star marks the position for the ideal catalyst with zero overpotential. Two-dimensional image of (**b**) dynamic tridimensional adsorption of the intermediates at the NiO/LDH intersection and (**c**) traditional single site adsorption on a planar surface. The yellow and green balls represent oxygen and hydrogen atoms, respectively. The red lines denote the hydrogen bonds. Reprinted from reference [113], copyright (2019), Wiley.

**Figure 12 materials-17-00199-f012:**
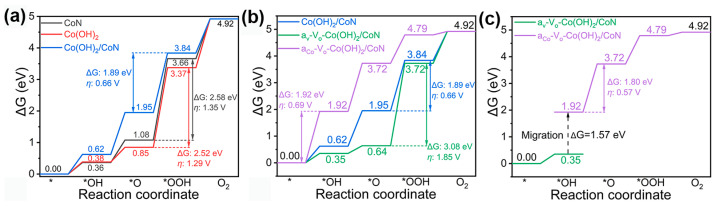
(**a**–**c**) Adsorption Gibbs free energy changes of OER intermediates on V_O_-Co(OH)_2_/CoN, Co(OH)_2_/CoN, CoN and Co(OH)_2_, where aCo-V_O_-Co(OH)_2_/CoN and a_V_-V_O_-Co(OH)_2_/CoN stand for the catalysts with active sites at Co and oxygen vacancy, respectively. Reprinted from reference [118], copyright (2023), Springer Nature.

**Table 1 materials-17-00199-t001:** Overpotentials and Tafel slopes of different HER catalysts.

Catalyst ^a^	Loading /mg cm^−2^	η_10_ /mV	Tafel Slope /mV dec^−1^	Ref.
β-Ni(OH)_2_/Pt	0.013	92	42	[48]
Pt-CoS_2_/CC ^b^	0.5	24	82	[49]
Ru_1_/def-NiFe LDH/NF ^c^	2	18	29	[52]
Ru_20_@ON-C ^d^	—	28	30	[80]
Ru/Ru_2_P	0.357	24	31.99	[41]
Ru/RuO_2_	—	34	29.01	[61]
cRu/Ni_3_N/NF	4	32	26.2	[50]
Co/CoP/C	0.88	193	73.8	[62]
Co/Ni_3_N/CC	2.91	194	156	[60]
Co/β-Mo_2_C@N-CNT	0.014	170	92	[58]
NiO/Ni/CNT	0.28	<100	82	[54]
Ni/CeO_2_/CNT	0.14	<100	—	[55]
Ni/Mo_2_C/C	0.5	60	52	[57]
Ni_3_N/Ni/NF	2.5	12	29.3/80.1	[63]
Ni/WC@N-C	0.7	77	68.6	[59]
Ni/CeO_2_@N-C_f_ ^e^	—	100	85.7	[56]
Cr-Ni_3_FeN/Ni@N-GTs ^f^	—	88	103	[81]
Ag-CoFe@N-C	—	260	66	[82]
MoP/MoS_2_/CF	—	58	59	[67]
S-MoS_2_@C	—	155	99	[66]
Mo_2_C/Mo_2_N	—	80	39.7	[28]
Mo_2_C/Mo_2_N/CC	5	54	48	[72]
Mo_5_N_6_/MoS_2_/HCNRs ^g^	1	53	37.9	[69]
(Ni_3_S_2_/MoS_2_)@TiO_2_/NF	—	49	38.9	[83]
1T-MoS_2_/Ni_3_S_4_/CC ^h^	2.18	44	43	[70]
Mo_2_C/MoO	—	115	54.3	[84]
MoP_2_/MoS_2_/NF	—	50	41	[68]
Co_2_P/Ni_2_P/NF	—	90	65.3	[75]
CoP/MXene	—	102	68.7	[85]
Co_2_P/CoP/N-C	—	91	86	[86]
Co_2_P-MXene-NF	27	42	75	[79]
CoP/Co_2_P/NP-C	—	109	78.9	[74]
Ni_x_P/Co_2_P/NF	—	170	112.92	[87]
Ni_2_P/Ni	—	84	114.2	[88]
NiP_2_/FeP_2_/Cu_nw_/CuF ^i^	—	23.6	52	[89]
HO-NiO/Cu	1.8	33	51	[53]
Mn-Ni_2_P/Fe_2_P	—	90	115.41	[90]
Ni_3_N/Ni_0.2_Mo_0.8_N/CC	—	79	86.8	[91]
ZnP@Ni_2_P/NiSe_2_/NF	6.5	79	82	[92]
Ni(OH)_2_/Fe-Ni_2_P/NF	—	45	43	[93]
NiSe_2_/NiMoN/NF	4.2	58	68.7	[37]
WO_v_N@NC/NF	1.5	16	33	[94]
CoSe/MoSe_2_	—	148	50	[95]
CoTe_2_/WTe_2_/NF	—	178	76	[96]

^a^: Except for the electrolyte for β-Ni(OH)_2_/Pt, which is 0.1 mol L^−1^ KOH, all other electrolytes are 1 mol L^−1^ KOH (pH = 14). ^b^: CC is carbon cloth. ^c^: NF is Ni foam. ^d^: ON-C is O, N-doped carbon. ^e^: C_f_ is carbon fiber. ^f^: GT is graphene tube. ^g^: HCNR is hollow carbon nanoribbons. ^h^: 1T-MoS_2_ is one of the three crystal structures of MoS_2_. 1T-MoS_2_ is metallic, in which the molybdenum atom in the monolayer MoS_2_ is octahedrally coordinated with the coordination number of 6. ^i^: Cu_nw_ is Cu nanowires. CuF is Cu foam.

**Table 2 materials-17-00199-t002:** Overpotentials and Tafel slopes of different OER catalysts.

Catalyst ^a^	Loading /mg cm^−2^	η_10_ (Other *J*) /mV (mA cm^−2^)	Tafel Slope /mV dec^−1^	Ref.
Ir/Co(OH)_2_	0.566	235	70.2	[98]
Ir/Ni(OH)_2_	0.2	224	41	[30]
Ir_1_/NiO	1.2	215	38	[99]
Ir/NiFeO	—	197	29.6	[100]
RuNiCoMOF/CoNiRu LDH	1.7	255 (20)	67	[103]
RuO_2_/NiO/NF	1.1	250	50.5	[101]
RuO_2_@Co_3_O_4_	—	152	68	[102]
a/c CoNiRuO_x_/N-C	—	245	82.3	[126]
FeNi LDH/rGO	0.25	~210	39	[110]
G/NiFeOH	—	310	39	[111]
NiFe@NiFeOOH	2.5	248 (500)	34.7	[112]
NiO/NiFe LDH/CuF	—	205 (30)	30	[113]
NiFe(OH)_x_/FeS/IF ^b^	5.2	245 (50)	—	[114]
Fe-NiO_x_/NF	—	215	34	[127]
NiCeO_x_/Au	—	271	—	[128]
Co_9_S_8_/Ni_3_S_2_	4	281 (50)	53.3	[117]
Co(O_v_H)_2_/CoN/CP	2	206	67	[118]
CoFeO_x_-mPCB/CC	0.928	232	32	[121]
Fe-Ni_3_S_2_/MnS/NF	—	259 (50)	39	[129]
Ni_3_S_2_@NiCo LDH/NF	—	305 (50)	47.5	[130]
Cu@CeO_2_@NiFeCr	0.2	230.8	32.7	[131]
Ni(OH)_2_/NiCo_2_O_4_/NF	3	224	41	[132]
hNiO/NF	—	380	299	[133]

^a^: Except for the electrolyte for NiCeO_x_/Au, which is 1 mol L^−1^ NaOH (pH = 14), all other electrolytes are 1 mol L^−1^ KOH (pH = 14). ^b^: IF is iron foam.

**Table 3 materials-17-00199-t003:** Overpotentials, Tafel slopes and cell voltages of different bifunctional catalysts.

Catalyst ^a^	HER		OER		U_10_ /V	Ref.
	η_10_ /mV	Tafel Slope /mVdec^−1^	η_10_ /mV	Tafel Slope /mV dec^−1^		
Co_3_O_4_@MoS_2_/CC	90	59.5	269	58	1.59	[124]
CoP/Co_9_S_8_	155	67	320	42	1.6	[134]
CoS_2_/MoS_2_/CC	71	62.8	274	57.5	1.59	[135]
Co_2_P/Co_3_O_4_/C	86	49.7	246	69.5	1.55	[125]
Fe-CoN/CoS_2_/CP	72	86	154	71	1.48	[136]
Ni_3_S_2_/MoS_2_	166	78.2	303	70.3	1.62	[137]
MoS_2_/Ni_3_S_2_/CA@CC	116	73	265	58	1.51	[138]
Ni_3_Fe_0.75_V_0.25_/Ni_3_Fe_0.75_V_0.25_N @NiFeOOH	113	56	260 (50)	77	1.66	[139]

^a^: All electrolytes are 1 mol L^−1^ KOH (pH = 14).

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
