# Peer review of "Nano-Scale Engineering of Heterojunction for Alkaline Water Electrolysis"

_materials, 2023, doi:10.3390/ma17010199_

Round 1
Reviewer 1 Report
Comments and Suggestions for Authors
In this Review, authors describe the state-of-the-art of heterojunction catalysts for alkaline HER and OER. After a theoretical introduction, the main properties of catalysts for alkaline HER, OER and based on heterojunction approach were presented and discussed and the difference with a single material were highlighted. Finally also bifunctional OER catalysts were presented and discussed. This review was well written, in a clear manner and each catalysts reported well described. However, before publication in Materials journal, a minor revision should be done.
My comment are reported below:
- On page 2, line 45, “…1 mol of hydrogen…” should be replaced with “…1 mol of water…” ;
- For a more clear introduction and also for a non-expert readers, Volmer-Tafel Volmer-Heyrovsky HER equations should be described in more details. Moreover, also a Sabadier principle should described more broadly;
- In Tables 1 2 and 3, the overpotentials reported at with current density were calculated?
- The pH conditions of the results reported for each catalysts should be added to the text;
- For the best catalysts of each class for both HER and OER, a more details about electrocatalysis such as electrocatalytic curves, test conditions , electrochemical characterizations should be added and commented to the main text.
Reviewer 2 Report
Comments and Suggestions for Authors
The submitted review collects recent advances on heterojunction for alkaline water electrolysis.
The review is well structured, well written and includes a chapter on mechanisms useful for the non-expert reader.
English language is fine
My opinion is to accept the submission and I suggest :
- authors should consider to add more keywords
- authors should consider to add a table of contents and a short list of abbreviations to help readers
- Colors of figure 10 do not have much contrast, especially the pink curve. While recognizing that it is reprinted from reference, authors should consider the possibility of improving the figure to optimize reading
Reviewer 3 Report
Comments and Suggestions for Authors
I appreciate dose of information in this study, which is clearly described and presented in the systematic way. I only suggest some minor corrections:
- The hyphenation between lines needs improvement.
- L 104 – add reference
- L 226 – introduce this symbol: εd , and explain it
- L 342 – index in OH-
- L 418,421,724 – indexes in chemical formulas
- L 422 – rGO – reduced graphite oxide
- L 468 – grammatic mistakes should be corrected
- L 469 – what was the role of dopamine?
- Table 1 – not everything is clear, please explain : HCNR, CC, N-C , 1T, CuNW/Cuf , please correct index in CoP/Co2P/NP-C
- L 561 – concentrated KOH?
- Fig 7 – please explain abbreviation of pds
Reviewer 4 Report
Comments and Suggestions for Authors
This review article presents heterogeneous materials for HER and OER electrocatalysis. My comments are as follows.
1. The author should add further relative works, https://doi.org/10.1007/s13391-023-00458-9, https://doi.org/10.1016/j.matlet.2022.132740, and discuss in detail.
2. Typo errors should be checked throughout the manuscript.
3. The author should describe the advantages of heterojunction for electrocatalysis application. 4. It is suggested to describe and discuss the methods for the fabrication of heterojunctions. 5. The author should mention future directions for the fabrication of heterojunctions for electrocatalysis applications.
